# Quantitative Prediction of Sea Clutter Power Based on Improved Grey Markov Model

**Zihao Chen** [1], **Bin Tian** [1,*], **Siyun Zhang** [1] and **Quanjun Xu** [2]

1   School of Electronic Engineering, Naval University of Engineering, Wuhan 430033, China;
    chen_lv0423@163.com (Z.C.); 18162626235@189.cn (S.Z.)
2   Naval Staff, Beijing 100841, China; lu17671099078@163.com
*   Correspondence: sweetybox123@163.com

**Abstract:** The detection and prediction of sea clutter power is the basis of inversing atmospheric duct. At present, the technology of atmospheric duct within radar detection range is relatively perfect, but the long-distance inversion of atmospheric duct is limited by radar detection range, and the prediction of the echo power of the measured sea clutter is the basis of long-distance inversion of atmospheric duct. Based on the theory of weighted Markov model and grey Markov model, a weighted grey Markov model is constructed, and the sliding method is introduced to establish the sliding weighted grey Markov model. The relative error between the measured sea clutter power and predicted values of the above four models is calculated and analyzed using the experimental data collected. The results show that the sliding weighted grey Markov model has better accuracy not only in short-range prediction but also in long-distance prediction, which could provide data support for inversing atmospheric duct.

**Keywords:** weighted Markov model; GM (1,1) model; improved grey Markov model; sea clutter power; refractivity from clutter

## 1. Introduction

Because radar sea clutter, in the propagation process, carries information on atmospheric refractive index and is affected by atmospheric factors, refractivity from clutter (RFC) uses the transmission characteristics of radar sea clutter in the atmospheric duct to obtain the atmospheric corrected refractive profile, which has become an important means of obtaining atmospheric duct [1,2]. Karimian [3] summarized the development status of RFC, Rogers [1] described a method for inferring the evaporation duct height from sea clutter using data from the Wallops' 98 measurement campaign, Gerstoft [4] gave the basic method of RFC and gave a method of inversion of horizontal range-dependent evaporation duct from radar sea clutter. Domestic scientific research institutes have carried out research on RFC [5–7]. Therefore, the detection of sea clutter echo power is the basis of inversing atmospheric duct. However, the research on RFC is limited by the radar detection distance, and only improves the inversion accuracy of range-dependent or non-atmospheric duct by perfecting the calculation method [8–11]. Therefore, the prediction of the echo power of the measured sea clutter is the basis of long-distance inversion of atmospheric duct, but there is a lack of research on the prediction of the measured sea clutter beyond radar detection distance.

The Markov model has higher prediction accuracy for discrete-time random data series, so it is the preferred method for many prediction problems. In recent years, many scholars have continuously improved the Markov model and established the weighted Markov model and the grey Markov model. At present, the Markov model is widely used in so many fields, such as communication, computer, meteorology, atmosphere, and other areas [12–17].

In view of the lack of research on long-distance atmospheric duct inversion, the echo power of sea clutter, in this paper, is predicted beyond the radar detection range, which lays a theoretical foundation for radar sea clutter inversion of long-distance atmospheric duct. The sea clutter power is a discrete random array sequence in the range direction, and the sea clutter power of the next position is only related to the power of the previous positions, therefore, it could be regarded as a stochastic process in discrete distance and the same properties as the discrete-time Markov process. However, the traditional grey Markov model simply takes the midpoint of the interval as the value in the quantitative solution, so the value has great randomness, and the array structure cannot be optimized in time by using the latest information of the array in the multi-step continuous prediction. In this paper, fuzzy set theory and sliding method are introduced to improve the grey Markov model, then, the improved grey Markov model is compared with the traditional grey Markov model to analyze the prediction accuracy, and a comparative analysis is given by using the experimental data collected. Finally, it comes to the conclusion that the improved grey Markov model has better prediction accuracy than traditional models and could provide data support for inversing atmospheric duct.

## 2. Markov Model and Fuzzy Set Theory

### 2.1. Markov Model

The Markov model is a common method to predict data series with no aftereffect, and its characteristics are that the future state is related to the current state, not affected by the past state, and it has high prediction accuracy. Therefore, it is the preferred method for most prediction problems.

The parameter set T of the Markov process, $\{X_n, n \in T\}$, is a discrete time set. However, for sea clutter power prediction, the detection distance of radar is relatively close, and the electromagnetic wave propagates at the speed of light, so it can be considered that sea clutter power on the same path is detected at the same time. What is more, sea clutter power is a stochastic sequence in the discrete range and has the characteristics that the sea clutter power of the latter position is only related to the several previous positions. Therefore, sea clutter power can be regarded as a stochastic process of discrete distance with Markov property, $\{X_n, n \in S\}$, and has the same properties as the discrete time Markov process.

### 2.2. Fuzzy Set Theory

When using the Markov model to predict, the influence of the present state on the latter state is usually considered, but several previous states may also affect the latter state. Therefore, the autocorrelation coefficient of each order is used as the influence weight of the several previous states on the latter state, and finally, the weighted sum of each state probability is used to predict the next state. The state level and its value range of the latter position can be predicted by the weighted Markov model, and the quantitative prediction can be calculated by using the level characteristics value in the fuzzy set theory.

#### 2.2.1. Calculation Method of State Level, Transition Matrix, and State Prediction

According to the statistical method, the mean value ($\bar{x}$)-standard deviation ($\sigma$) method is used to grade the measured values, and the sample is divided into five states, as shown in Table 1.

**Table 1.** State classification standard of mean value ($\bar{x}$)-standard ($\sigma$) deviation method.

| State Level | Interval Range |
| --- | --- |
| 1 | $x < \bar{x} - \sigma$ |
| 2 | $\bar{x} - \sigma \leq x < \bar{x} - 0.5\sigma$ |
| 3 | $\bar{x} - 0.5\sigma \leq x < \bar{x} + 0.5\sigma$ |
| 4 | $\bar{x} + 0.5\sigma \leq x < \bar{x} + \sigma$ |
| 5 | $\bar{x} + \sigma \leq x$ |

According to the frequency, the transition probability can be calculated, assuming that $f_{ij}^1$ is the frequency of the measured value from state i to state j in one step, the one-step transition probability can be expressed as Equation (1).

$$p_{ij}^1 = f_{ij}^1 / \sum_{j=1}^{m} f_{ij}^1 \tag{1}$$

One-step transition matrix P and n steps transition matrix P^(n) can be obtained from Equation (1).

According to the full probability formula, the state of the measured value at position s is i, then the probability vector of the predicted value that is at state j after k steps transfer is Equation (2).

$$a_{(s+k)j} = \sum_{i=1}^{m} a_{si} p_{ij}^k \tag{2}$$

where m is the total number of states, $a_{si}$ is the probability that measured value is state i at position s.

The state of the predicted value at position $s_p$ can be calculated due to $p_{max}(s_p) = max(p_i)$.

### 2.2.2. Quantitative Prediction Based on Fuzzy Set Theory

According to fuzzy set theory, the K-order correlation coefficient could be expressed as Equation (3).

$$r_K = \sum_{i=1}^{N-K} (x_i - \bar{x})(x_{i+K} - \bar{x}) / \sum_{i=1}^{N-K} (x_i - \bar{x})^2 \tag{3}$$

The weight factor could be expressed as Equation (4).

$$\omega_K = |r_K| / \sum_{K=1}^{n} |r_K| \tag{4}$$

where N is the number of samples, n is the total order of correlation coefficient.

The state of the predicted value is the position corresponding to the largest element after the weighted average of probability vectors with different steps. What is more, the weight set, $D = \{d_1, d_2, \cdots, d_m\}$, composed of the weights of each state can be obtained, $d_i$ could be expressed as Equation (5).

$$d_i = p_i^\eta / \sum_{i=1}^{m} p_i^\eta \tag{5}$$

where i is the state level, $p_i$ is the probability that the predicted value is state i, $\eta$ is the maximum probability action coefficient, usually taken as 2 [18].

State eigenvalue H can be expressed as Equation (6)

$$H = \sum_{i=1}^{m} i d_i \tag{6}$$

According to [19], the method of quantitative prediction using state eigenvalues is as Equation (7).

$$X_{prediction} = \begin{cases} T_i H / (i + 0.5), H > i \\ B_i H / (i - 0.5), H < i \end{cases} \tag{7}$$

where i is the state determined according to the maximum probability, $T_i$ and $B_i$ are the upper and lower limits of the corresponding interval of state i.

## 3. Grey Markov Model

### 3.1. GM (1,1) Model

The main principle of the grey model is to obtain a group of data, $\{X_1(n), n \in S\}$, with certain regularity from the irregular original sample data, $\{X_0(n), n \in S\}$ by accumulation or subtraction method, and accumulation is selected as Equation (8).

$$\begin{cases} x_1(1) = x_0(1) \\ x_1(i) = x_0(1) + x_0(2) + \cdots x_0(i) \end{cases} \tag{8}$$

and construct matrix **B** and vector $\vec{y_n}$. The expression is as Equation (9).

$$\mathbf{B} = \begin{bmatrix} -\frac{1}{2}(x_1(1) + x_1(2)) & 1 \\ -\frac{1}{2}(x_1(2) + x_1(3)) & 1 \\ \cdots & \cdots \\ -\frac{1}{2}(x_1(n-1) + x_1(n)) & 1 \end{bmatrix}, \vec{y_n} = \begin{bmatrix} x_0(2) \\ x_0(3) \\ \cdots \\ x_0(n) \end{bmatrix} \tag{9}$$

The modeling process of the grey system adopts a differential fitting method, GM (1,1) model is widely used. The specific process is as Equation (10):

$$\frac{dx_1(s)}{dt} + ax_1(s) = u \tag{10}$$

The parameter u is the grey action quantity, and it can reflect the relationship of change data, whose exact meaning is grey. The parameter a is the development coefficient, and it is related to GM (1,1). The relationship is shown in Table 2.

**Table 2.** The relationship of development coefficient a and applicable scope of GM (1,1) model.

|   | Development Coefficient a | Applicable Scope of GM (1,1) Model |
|---|---|---|
| 1 | $-0.3 \leq a$ | Applicable to medium and long term |
| 2 | $-0.5 \leq a < -0.3$ | Applicable to short term |
| 3 | $-0.8 \leq a < -0.5$ | Suitable for short term, but with caution |
| 4 | $-1 \leq a < -0.8$ | Residual correction |
| 5 | $a < -1$ | Not suitable for use |

$\vec{a} = [a, u]^T$ is called coefficient vector, and it can be solved by the least square method. The result is expressed as Equation (11).

$$\vec{a} = \left(B^T B\right)^{-1} B^T \vec{y_n} \tag{11}$$

Then, the response sequence of GM (1,1) model is expressed as Equation (12).

$$\hat{x}_1(k+1) = \left(x_0(1) - \frac{u}{a}\right) e^{-ak} + \frac{u}{a}; k = 1, 2, \tag{12}$$

The fitting value of the original sample data is expressed as Equation (13).

$$\hat{x}_0(k+1) = \hat{x}_1(k+1) - \hat{x}_1(k) = (1 - e^a)\left(x_0(1) - \frac{u}{a}\right) e^{-ak}; k = 1, 2, \tag{13}$$

### 3.2. Grey Markov Model

$\{x_0(n), n \in S\}$ is original sample data, and $\hat{x}_0(s)$ is the fitting value of the original data, $x_0(s)$, at position s. According to GM (1,1), $\hat{x}_0(k)$ is the predicted value of the fitting value at position k. The specific modeling method is as follows:

According to residual relative value, $q = (\hat{x}_0(s) - x_0(s))/x_0(s)$, the state level is divided as shown in Table 3, where $q_{min}$ is the minimum of residual relative value, $q_{max}$ is the maximum.

**Table 3.** State level division.

| State | Interval Range |
|---|---|
| 1 | $q_{min} \le q < 4/5q_{min} + 1/5q_{max}$ |
| 2 | $4/5q_{min} + 1/5q_{max} \le q < 3/5q_{min} + 2/5q_{max}$ |
| 3 | $3/5q_{min} + 2/5q_{max} \le q < 2/5q_{min} + 3/5q_{max}$ |
| 4 | $2/5q_{min} + 3/5q_{max} \le q < 1/5q_{min} + 4/5q_{max}$ |
| 5 | $1/5q_{min} + 4/5q_{max} \le q < q_{max}$ |

According to Section 2.2.1, the 1–5 step state transition probability matrix, $\mathbf{P}, \mathbf{P}^{(2)}, \cdots, \mathbf{P}^{(5)}$, can be calculated, and the predicted state can be obtained. The result is expressed as shown in Table 4.

**Table 4.** State prediction of residual relative value.

| Initial Position | Transfer Steps | Initial State | State 1 | State 2 | State 3 | State 4 | State 5 |
|---|---|---|---|---|---|---|---|
| $i_1$ | 1 | $m_1$ | $p^1_{m_1 1}$ | $p^1_{m_1 2}$ | $p^1_{m_1 3}$ | $p^1_{m_1 4}$ | $p^1_{m_1 5}$ |
| $i_2$ | 2 | $m_2$ | $p^2_{m_1 1}$ | $p^2_{m_1 2}$ | $p^2_{m_1 3}$ | $p^2_{m_1 4}$ | $p^2_{m_1 5}$ |
| $i_3$ | 3 | $m_3$ | $p^3_{m_1 1}$ | $p^3_{m_1 2}$ | $p^3_{m_1 3}$ | $p^3_{m_1 4}$ | $p^3_{m_1 5}$ |
| $i_4$ | 4 | $m_4$ | $p^4_{m_1 1}$ | $p^4_{m_1 2}$ | $p^4_{m_1 3}$ | $p^4_{m_1 4}$ | $p^4_{m_1 5}$ |
| $i_5$ | 5 | $m_5$ | $p^5_{m_1 1}$ | $p^5_{m_1 2}$ | $p^5_{m_1 3}$ | $p^5_{m_1 4}$ | $p^5_{m_1 5}$ |
| Total | - | - | $P_1$ | $P_2$ | $P_3$ | $P_4$ | $P_5$ |

From Table 4, $p_{max}(k) = max(p_i), i = 1, 2, \cdots, 5$ can be solved. Assuming that the probability of state i is the largest, then the residual relative value is in the corresponding interval of state i, so the predicted residual relative value is most likely the midpoint of the interval. From GM (1,1) model, the fitting value, $\hat{x}_0(k)$, could be calculated, then according to the definition of residual relative value, the predicted value of the original sample at position k, $x_0(k)$, can be expressed as Equation (14).

$$x_0(k) = \hat{x}_0(k) / (1 + \frac{1}{2}(B_i + T_i)) \tag{14}$$

where i is the state determined according to the maximum probability, $T_i$ and $B_i$ are the upper and lower limits of the corresponding interval of state i.

## 4. Improved Grey Markov Model

However, the traditional grey Markov model simply takes the midpoint of the interval as the value in the quantitative solution, so the value has great randomness, and the array structure cannot be optimized in time by using the latest information of the array in the multi-step continuous prediction. Therefore, fuzzy set theory and sliding method are introduced to improve the grey Markov model.

### 4.1. Weighted Grey Markov Model

On the basis of GM (1,1) model in Section 2, fuzzy set theory is used to quantitatively predict residual relative value at position k, instead of the midpoint. According to the weighted Markov model,

The K-order correlation coefficient and weight factor can be expressed as Equations (4) and (15).

$$r_K = \sum_{i=1}^{N-K}(q_i - \bar{q})(q_{i+K} - \bar{q}) / \sum_{i=1}^{N-K}(q_i - \bar{q})^2 \tag{15}$$

where N is the number of samples, n is the total order of correlation coefficient.

Then, according to Equations (5)–(7), the quantitative result of residual relative value can be calculated, from Equation (14), the predicted value of the original sample at position k, $x_0(k)$, can be obtained.

*4.2. Sliding Weighted Grey Markov Model*

When predicting the sea clutter power, $\{X_{n'}, n' \in S\}$ at a long distance, $n' = 1, 2, 3, \cdots$, $n + 1$, $n + 2$, $n + 3, \cdots$, with the increase of the number of sample, the impact of the remote original data on the prediction will continue to decrease, so they would lose reference significance. In this way, the remote data should be constantly deleted from the original data, and new predicted values should be added to form a new research system, then prediction could be continuously and accurately carried out. As an improved model of the weighted grey Markov model, the sliding method is to optimize the system structure by using the latest predicted information. Specifically, at first, assuming that the original sample is $x_0 = \{x_0(1), x_0(2), \cdots, x_0(n)\}$, the predicted value at position n + 1, $x_0(n + 1)$, can be calculated from the weighted grey Markov model, secondly, the remote data $x_0(1)$ would be deleted, and the latest data, $x_0(n + 1)$, would be added, then a new sample, $x_0' = \{x_0(2), x_0(3), \cdots, x_0(n + 1)\}$ could be obtained, then $x_0'' = \{x_0(3), x_0(4), \cdots, x_0(n + 2)\}$, according to the model, $x_0(n + 3), x_0(n + 4), \cdots$, could be calculated.

## 5. Application of Models

In this paper, a certain type of magnetic control pulse sea detection radar was used, and the radar was set up along the coast of a certain sea area in the Yellow Sea and the Bohai Sea in early December 2021. The result of sea clutter power is shown in Figure 1 where *x*-axis and *y*-axis are the numbers of positions detected by radar.

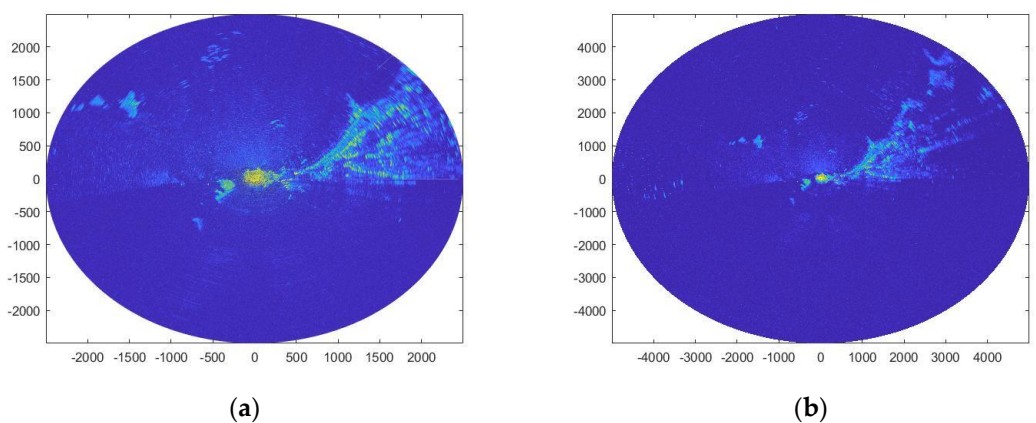

(**a**)　　　　　　　　　　　　　　　　　　　　　　　(**b**)

**Figure 1.** The measured sea clutter power. (**a**) The echo power of sea clutter within 6 km, (**b**) the echo power of sea clutter within 6 km.

In Figure 1, the radar was set up at center of circle that is axis origin, and it can detect sea clutter in all directions. The brightness of the circular area in the Figure 1 reflects the intensity of the sea clutter echo power. The brighter the area is, the higher the intensity is. The detection range of the radar is reflected by the values of *x*-axis and *y*-axis. Specifically, the value of axis is the detection position of the radar, and one position is 2.4 m, so Figure 1a shows the echo power of sea clutter within 6 km and Figure 1b is 12 km. It can be seen that in North-East in Figure 1, there was atmospheric duct. What is more, from Figure 1, we can see the sea clutter power of later positions is related to the power of the previous positions, and the echo power of sea clutter decreases with the increase of propagation distance.

A group of the same samples is randomly selected, and the weighted Markov model, grey Markov model, the weighted grey Markov model, and the sliding weighted grey Markov model are used to predict sea clutter power at the latter position and in long distance.

### 5.1. Weighted Markov Model

According to the mean value ($\bar{x}$)-standard deviation ($\sigma$) method, sea clutter power at the last 200 positions of the sample is divided into five states as shown in Table 5, where $\bar{x} = 4.45$ and $\sigma = 0.38$.

**Table 5.** State division of sea clutter power.

| State Level | Interval Range |
|:---:|:---:|
| 1 | x < 4.07 |
| 2 | 4.07 ≤ x < 4.26 |
| 3 | 4.26 ≤ x < 4.65 |
| 4 | 4.65 ≤ x < 4.83 |
| 5 | 4.83 ≤ x |

x is the amplitude value of dimensionless units, and 0 linear corresponds to 0 V, and 255 linear corresponds to 8 V.

One-step transition matrix **P**, n steps transition matrix $\mathbf{P}^{(n)}$, k-order correlation coefficient $r_k$ and weight factor $\omega_k$ can be calculated, as shown in Table 6 and Equation (16).

$$\begin{cases} \mathbf{P} = \mathbf{P}^{(1)} = \begin{bmatrix} 0.58 & 0.28 & 0.14 & 0.0000 & 0.0000 \\ 0.30 & 0.26 & 0.44 & 0.0000 & 0.0000 \\ 0.09 & 0.13 & 0.58 & 0.15 & 0.05 \\ 0.0000 & 0.0000 & 0.55 & 0.27 & 0.18 \\ 0.0000 & 0.0000 & 0.09 & 0.11 & 0.80 \end{bmatrix} \\ \mathbf{P}^{(n)} = \mathbf{P}^n \end{cases} \tag{16}$$

**Table 6.** Correlation coefficient and weight factor of each order.

| Order | 1 | 2 | 3 | 4 | 5 |
|:---:|:---:|:---:|:---:|:---:|:---:|
| $r_K$ | 0.82 | 0.55 | 0.39 | 0.31 | 0.26 |
| $\omega_K$ | 0.35 | 0.24 | 0.17 | 0.13 | 0.11 |

The state of the sea clutter power of the latter position can be predicted from the states of the last five positions of the sample, as shown in Table 7.

**Table 7.** State prediction of the sea clutter power of the latter position.

| Initial Position | Transfer Steps | Initial State | $\omega_K$ | State 1 | State 2 | State 3 | State 4 | State 5 |
|:---:|:---:|:---:|:---:|:---:|:---:|:---:|:---:|:---:|
| 200 | 1 | 5 | 0.35 | 0 | 0 | 0.09 | 0.11 | 0.80 |
| 199 | 2 | 4 | 0.24 | 0.05 | 0.07 | 0.48 | 0.18 | 0.22 |
| 198 | 3 | 3 | 0.17 | 0.16 | 0.14 | 0.45 | 0.12 | 0.13 |
| 197 | 4 | 2 | 0.13 | 0.25 | 0.17 | 0.41 | 0.09 | 0.08 |
| 196 | 5 | 1 | 0.11 | 0.27 | 0.18 | 0.40 | 0.08 | 0.07 |
| Total | - | - | | 0.10 | 0.08 | 0.32 | 0.13 | 0.37 |

From Table 7 and Equations (5) and (6), the state eigenvalue, H, can be calculated, then the sea clutter power of the latter position can be obtained, as shown in Table 8.

**Table 8.** Quantitative prediction of the sea clutter power of the latter position.

| State | Interval Range | H | Predicted | Measured | Relative Error |
|:---:|:---:|:---:|:---:|:---:|:---:|
| 5 | (4.83, ∞) | 3.98 | 4.27 | 4.90 | 12.9% |

### 5.2. Grey Markov Model

According to the GM (1,1) model, the fitting value of sea clutter power at the last 200 positions of the sample can be calculated, then, from the grey Markov model, the sea clutter power of the latter position can be obtained as shown in Tables 9–11.

**Table 9.** State level division of residual relative value (q).

| State Level | Interval Range |
|:---:|:---:|
| 1 | $-0.17 \leq q < -0.08$ |
| 2 | $-0.08 \leq q < 0$ |
| 3 | $0 \leq q < 0.08$ |
| 4 | $0.08 \leq q < 0.17$ |
| 5 | $0.17 \leq q < 0.25$ |

q is a parameter of dimensionless units.

**Table 10.** State prediction of residual relative value by the grey Markov model.

| Initial Position | Transfer Steps | Initial State | State 1 | State 2 | State 3 | State 4 | State 5 |
|:---:|:---:|:---:|:---:|:---:|:---:|:---:|:---:|
| 200 | 1 | 1 | 0.57 | 0.43 | 0 | 0 | 0 |
| 199 | 2 | 1 | 0.16 | 0.48 | 0.29 | 0.06 | 0.01 |
| 198 | 3 | 2 | 0.16 | 0.44 | 0.29 | 0.09 | 0.02 |
| 197 | 4 | 3 | 0.10 | 0.36 | 0.33 | 0.17 | 0.04 |
| 196 | 5 | 4 | 0.07 | 0.29 | 0.35 | 0.23 | 0.06 |
| Total | - | - | 1.0686 | 2.0038 | 1.2521 | 0.5488 | 0.1266 |

**Table 11.** Quantitative prediction of the sea clutter power of the latter position by the grey Markov model.

| Fitting Value | State | Interval Range | Midpoint | Predicted | Measured | Relative Error |
|:---:|:---:|:---:|:---:|:---:|:---:|:---:|
| 4.29 | 2 | $(-0.08, 0)$ | $-0.04$ | 4.47 | 4.90 | 8.8% |

### 5.3. Improved Grey Markov Model

Two improved grey Markov models, the weighted grey Markov model and the sliding weighted grey Markov model, have the same result when only predicting the sea clutter power of the latter position, but when predicting for long distance, they have significant differences.

According to Section 3, the sea clutter power of the latter position can be obtained as shown in Tables 12–14.

**Table 12.** Correlation coefficient and weight factor of each order of the improved grey Markov model.

| Order | 1 | 2 | 3 | 4 | 5 |
|:---:|:---:|:---:|:---:|:---:|:---:|
| $r^q_K$ | 0.79 | 0.49 | 0.31 | 0.22 | 0.16 |
| $\omega^q_K$ | 0.40 | 0.25 | 0.16 | 0.11 | 0.08 |

**Table 13.** State prediction of residual relative value by the improved grey Markov model.

| Initial Position | Transfer Steps | Initial State | $\omega_K$ | State 1 | State 2 | State 3 | State 4 | State 5 |
|:---:|:---:|:---:|:---:|:---:|:---:|:---:|:---:|:---:|
| 200 | 1 | 1 | 0.40 | 0.57 | 0.43 | 0 | 0 | 0 |
| 199 | 2 | 2 | 0.25 | 0.16 | 0.48 | 0.29 | 0.06 | 0.01 |
| 198 | 3 | 2 | 0.16 | 0.16 | 0.44 | 0.29 | 0.09 | 0.02 |
| 197 | 4 | 3 | 0.11 | 0.10 | 0.36 | 0.33 | 0.17 | 0.04 |
| 196 | 5 | 4 | 0.08 | 0.07 | 0.29 | 0.35 | 0.23 | 0.06 |
| Total | - | - | | 0.31 | 0.43 | 0.18 | 0.07 | 0.01 |

**Table 14.** Quantitative prediction of the sea clutter power of the latter position by the improved grey Markov model.

| Fitting Value | State | Interval Range | H | q | Predicted | Measured | Relative Error |
|---|---|---|---|---|---|---|---|
| 4.29 | 2 | $(-0.08, 0)$ | 1.83 | $-0.10$ | 4.78 | 4.90 | 2.53% |

Through the analysis of the above models, it can be seen that the relative error of the improved grey Markov model in predicting the latter position is reduced from 12.9% to 2.53%, indicating that the improved grey Markov model can improve the prediction accuracy. Next, the above four models are used to predict the long-distance sea clutter power of the same sample, and the prediction accuracy of each model will be compared. The results are shown as Table 15.

**Table 15.** Comparison of prediction accuracy of four models.

| Model Name | Predicted Positions | Average Relative Error |
|---|---|---|
| Weighted Markov model | 500 | 10.1% |
| Grey Markov model | 500 | 9.5% |
| Weighted grey Markov model | 500 | 12.3% |
| Sliding weighted grey Markov model | 500 | 8.6% |

*5.4. Model Applicability Analysis*

Only one sample has been analyzed in the first three sections of Chapter 4, so there is large randomness. In order to better illustrate the applicability of the sliding weighted grey Markov model, all the experimental data are calculated and analyzed, and the result is shown as Figure 2.

From Figure 2, the results are as follows:

- First, the grey Markov model has better accuracy in long-distance prediction than the weighted Markov model. The reason is that the fitting value is calculated and predicted, and the fitting value is closer to the sample.
- Second, in most cases, the weighted grey Markov model is better than the grey Markov model. The reason is that residual relative value can be estimated more accurately and quantitatively by introducing the correlation coefficient and weight factor of each order. However, in a few cases, the midpoint of the interval is the better prediction of residual relative value, thus the grey Markov model is better.
- Third, among four models, the sliding weighted Markov model has the best accuracy not only for the latter point but also for long-distance prediction.
- Fourth, it can also be seen that there are some large relative errors. Through the analysis of the measured data, it is found that the sea clutter power increased abnormally in several tests of large relative error, which is judged as the discovery of targets. After deleting positions of the target power, the relative error is greatly reduced, therefore, this model should be used to predict sea clutter power.

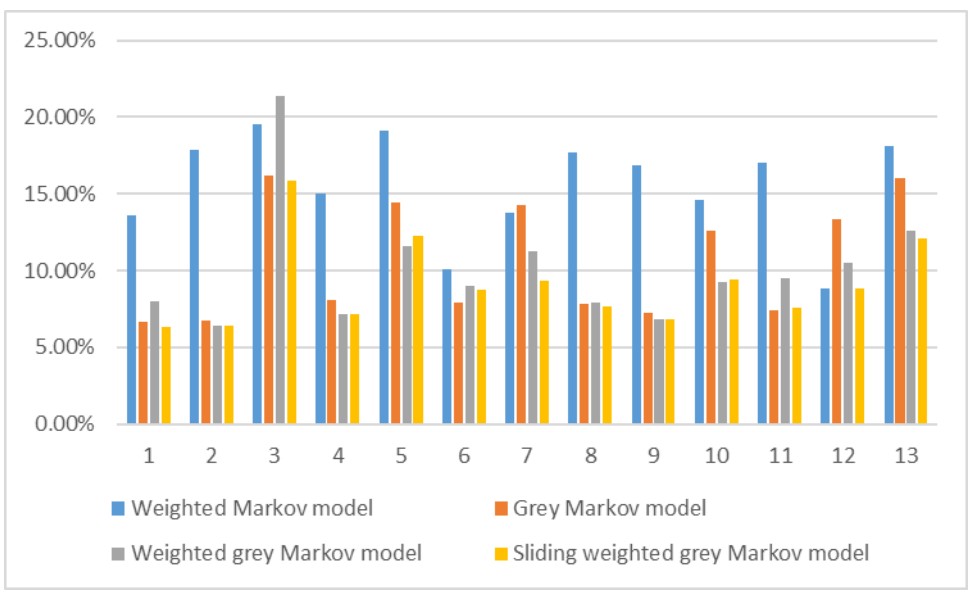

**Figure 2.** Average relative error of four models.

**6. Conclusions**

By analyzing the weighted Markov model and GM (1,1) model and improving the grey Markov model, the sliding weighted grey Markov model is obtained. What is more, the accuracy of the model is analyzed by using the measured sea clutter power. The results show that the sliding weighted grey Markov model has better accuracy for the prediction of short or long distances. Next, on the basis of RFC, the sliding weighted grey Markov model will be used to test whether the atmospheric duct from the predicted sea clutter power could reflect the real atmospheric environment information.

**Author Contributions:** Conceptualization, Z.C. and B.T.; methodology, Z.C. and B.T.; software, Z.C. and B.T.; validation, Z.C. and B.T.; formal analysis, Z.C. and B.T.; data curation, Z.C. and B.T.; writing—original draft preparation, Z.C. and B.T.; writing-review and editing, Z.C., B.T., S.Z. and Q.X. All authors have read and agreed to the published version of the manuscript.

**Funding:** This paper is sponsored by National Natural Science Foundation of China, 41975005.

**Institutional Review Board Statement:** Not applicable.

**Informed Consent Statement:** Not applicable.

**Data Availability Statement:** Data available on request due to restrictions eg privacy. The data presented in this study are available on request from the corresponding author.

**Conflicts of Interest:** The authors declare no conflict of interest.

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
