# Peer review of "Quantitative Prediction of Sea Clutter Power Based on Improved Grey Markov Model"

_atmosphere, doi:10.3390/atmos13071085_

Round 1

Reviewer 1 Report

1, The Figure name and Table name must be the same with that described in text

Author Response

Dear reviewer,

    Hope you are doing well. Thank you very much for your letter and advice. We have revised the paper, and would like to re-submit it for your consideration. We have addressed the comments raised by the reviewers, and the amendments are marked up using the “Track Changes” function in the revised manuscript. And for your comments, I have also revised the manuscript.

     We hope that the revision is acceptable, and I look forward to hearing from you soon.

With best wishes,

Yours sincerely,

Zihao Chen

Reviewer 2 Report

The article deals with the issues of quantitative prediction of sea clutter power based on improved grey Markov model. A brief introduction is given. It deals with only 7 sources. And the list of references includes only 12 sources. I did not find all the sources mentioned in the text. I think the list of references should be expanded including more newer sources.

The second section describes the models and fuzzy set theory, the third section describes the improved grey Markov model. The models are described in sufficient detail. Application of models is presented in fourth section. The Figure 1 is not informative. I recommend that the authors present this Figure in a form that allows readers to fully understand what authors want to show. The results of the research are summarized in sufficient detail. The conclusions could be more comprehensive.

Author Response

Dear reviewer,

Hope you are doing well. Thank you very much for your advice. We have revised the paper, and would like to re-submit it for your consideration. We have addressed your comments, and the amendments are marked up using the “Track Changes” function in the revised manuscript. 

We hope that the revision is acceptable, and I look forward to hearing from you soon.

With best wishes,

Yours sincerely,

Zihao Chen

Reviewer 3 Report

The authors address the sea clutter and the atmospheric duct for radar images registered on sea surface with significant waves or bad weather conditions.

The main drawback of the paper is that it is not obvious that these problems are effectively set and solved. Parameters and variables must be clearly stated and in this text these information are missing. Consequently, we suggest that the paper will be rejected. Here follows some examples.

Lines 87, 89 and 91 give three interpretations of k: a step number, a position, and a correlation coefficient order. s is equally a position. Check k in eq (4) and “i” in eq (5).

The parameter a is set in eq (9) and the opposite –a is presented in the text.

“the original” line 129

“sea clutter power” and “long distance” line 168 must correspond to a function or a variable defined before.

Check prime in x_0’ line 178

Check why the lines and several columns of the transition matrix P has a summation equal to 1 and columns 3 and 4 not.

Equations line 202

Give units of q in table 9 and x in table 5 and of axis, more function in fig 1.

Tables 8 and 11 could have the same presentation

“previous paper” could be  edited line 232

Distances line 259

Edit ref 2

Author Response

Dear reviewer,

    Hope you are doing well. Thank you for your detailed comments on my paper. We have revised the paper, and would like to re-submit it for your consideration. We addressed your comments  point by point, and the amendments are marked up using the "Track Changes" function in the revised manuscript.  Please see the attachment.

    We hope that the revision is acceptable, and I look forward to hearing from you soon.

With best wishes,

Yours sincerely,

Zihao Chen

Round 2

Reviewer 3 Report

Thanks to the resubmission of the paper, and the response letter to reviewer, we give here after some comments and sources of improvement.

Are clearer:

-          The word “duct” appeared and still appear 14 times in the abstract and in the introduction and one time in the conclusion. Thanks to the authors that atmospheric duct is not the object of the paper.

-          The data used seems to be “horizon free” there are no ship no island in the data used. We suggest mentioning the hypothesis that the research is not based on data where an object is present.

-          The research focusses only on the sea clutter power SCP on long-distance. We understand that paper will only present a comparison with short distance in fig(1).

Is less clear or remains unclear:

-          The “i” in eq(5), idem “k” in eq(4)

-          The paper uses several methods and introduces several notations, rendering difficult to think what results in term of Markov process.

-          The authors use 200 positions but present only one translation x_0(1) to x_0’. Is n=200 in section 5.1?

Is now becoming unclear:

-          Why the last row of matrix P in eq(16) is the first line of Table 7 ?

-          Why level 1 in Table (1°) is mean(x) - sigma and it is 4.07 in Table 5 with mean 3.9841 and sigma 0.5299?

-          Why title of rows 3 and 4 in Table 15 are different from subsection numbers?

NB1: If the SCP occurs on North-East in fig 1, the authors could mention the atmospheric or sea state origin.

NB2: Is eq(12) an inversion of eq(11)?

Conclusion: we suggest asking for a revision of the paper taking into account to the above observations.

Here we give some typos.

) eq 1

Accumulation or subtraction method may be edited, line 120

Eq(15) is eq(4)

Author Response

Dear reviewer,

    Hope you are doing well. We would like to thank you for the opportunity to revise and resubmit our manuscript. We found the your comments to be helpful in revising the manuscript and have carefully considered and responded to each suggestion, corresponding changes to the resubmitted manuscript are marked up using the “Track Changes”. Please see the attachment.

With best wishes,

Yours sincerely,

Zihao Chen
